# Histological Changes in Portal Cavernoma Cholangiopathy

**DOI:** 10.3390/diagnostics13030436

**Published:** 2023-01-25

**Authors:** Archana Rastogi, Chhagan Bihari, Shalini L. Thapar, Vikram Bhatia

**Affiliations:** Departments of Pathology Radiodiagnosis Hepatology, Institute of Liver and Biliary Sciences, D1, Vasant Kunj, New Delhi 110070, India

**Keywords:** portal biliopathy, portal cavernoma, liver biopsy, portal vein, bile ducts

## Abstract

Introduction: Portal cavernoma cholangiopathy (PCC)’ refers to abnormalities of the extrahepatic and intrahepatic bile ducts in patients with portal hypertension. Although there is data on clinical and imaging aspects of PCC, the description of liver pathology has been strikingly deficient. The purpose of this study was to examine the histopathological characteristics of PCC. Patients and Methods: A retrospective study of patients clinically diagnosed with extrahepatic portal vein obstruction (EHPVO) with portal cavernoma cholangiopathy, was conducted. Vascular anatomy was characterized by computerized tomographic angiography, and endoscopic retrograde cholangiography (ERC) and magnetic resonance cholangiography (MRC) were used to characterize the biliary anatomy. Histological features were analyzed by two hepatopathologists in a blinded manner, with mutual discussion to resolve any discrepancies. Results: A total of 50 patients with portal cavernoma cholangiopathy were included in the study. The mean age of the patients was 26.2 ± 11.6 years. Radiologically, bilobar intrahepatic biliary dilatation was seen in 98% with common bile duct abnormality in 100% of patients, along with extrinsic ductal impressions in 77 % of cases. Liver tests were deranged total bilirubin 1.5 mg/dL (IQR 0.8–2.4) and alkaline phosphatase 109.5 IU/L (IQR 70–193). Histologically; dilated multiple portal venous channels (72%), hepatic artery thickening (70%). The presence of aberrant vascular channels around portal tracts (54%), elastosis of portal veins (50%), and bile ductular reaction in (44%) were the other prominent findings. A 12% of cases show focal thin bridges. Advanced fibrosis was not seen in any of the cases. One-fourth of the cases showed concomitant minimal to mild hepatocyte steatosis. Conclusions: Histologically, intrahepatic portal vein and portal tract abnormalities were noted in cases with portal cavernoma cholangiopathy, associated with mild derangement of liver tests.

## 1. Introduction

The term “Portal cavernoma cholangiopathy (PCC)” refers to abnormalities of the extrahepatic and intrahepatic bile ducts in patients with long-standing portal hypertension (PHT) [1], usually due to extrahepatic portal vein obstruction (EHPVO) [2,3,4,5]. Cholangiography abnormalities in PCC include biliary indentations by paracholedochal collaterals, localized strictures, bile duct angulation or displacement, and ductal ectasias [1,6].

EHPVO may develop in the absence of cirrhosis and hepatocellular carcinoma and is one of the major causes of portal hypertension [7]. In EHPVO the pre-hepatic branches are involved with or without the involvement of the intrahepatic branches of PV, splenic vein, or superior mesenteric veins [8,9]. The significance of EHPVO lies in its secondary manifestation in the form of Portal biliopathy. Prompt diagnosis and treatment of portal cavernoma cholangiopathy are vital because chronic obstruction can lead to cholangitis or secondary biliary cirrhosis [10,11,12].

Portal cavernoma cholangiopathy is reported in 70%–100% of patients with portal cavernoma [13] and is often asymptomatic [14]. However, it can progress to more advanced stages with cholestasis, jaundice, biliary sludge, gallstones, cholangitis, and biliary cirrhosis [15]. 

The diagnosis of PCC is based on the combination of serum chemistry, ultrasound with color doppler imaging, biliary tract imaging based on magnetic resonance imaging (MRI) with MR cholangiopancreatography (MRCP) or endoscopic retrograde cholangiography (ERCP), and magnetic resonance portovenography [6,15].

A liver biopsy is not required for the diagnosis. However, a liver biopsy may be done by surgeons to rule out cirrhosis before performing shunt surgery, or concurrently with the shunt surgery [16,17]. The Asian Pacific Association for the Study of the Liver (APASL) consensus on EHPVO suggests that while all patients with EHPVO do not need a liver biopsy, it is helpful in the assessment of hepatic parenchymal injury [1]. Shunt surgeries are performed in specialized tertiary care centers. A significant cohort of cases motivated us to analyze histology in detail, where only scarce literature is available. Knowledge of the spectrum of histological changes is could be an important addition to the literature as a reference for pathologists and clinicians.

So, this study aimed to assess the possible histopathological changes in liver parenchyma in the cases of portal cavernoma cholangiopathy

## 2. Patients and Methods

### 2.1. Patients

This was a single-center study conducted at the Institute of Liver and Biliary Sciences (ILBS), New Delhi, India. We retrospectively analyzed patients with EHPVO and PCC who were evaluated between Jan 2012 and Dec 2019. A CT angiography was the study of choice for characterizing extra hepatic portal vein anomalies. ERCP and magnetic resonance cholangiography were used in the evaluation of biliary tract anomalies. A dynamic triple-phase multi-dissection computerized tomography (MDCT) study was performed on a 64-row spectral CT scanner. MRCP imaging was performed on a 3 Tesla scanner using a phased array TORSOPA coil. Sixty cases with clinically and radiologically proven portal cavernoma cholangiography who had complete clinical details recorded in 8 years (January 2012–December 2019) were found.

The inclusion criterion consisted of patients with EHPVO with overlying features of PCC. The triad used to make a diagnosis is the presence of portal cavernoma cholangiography changes on ERCP or MRCP, and the absence of alternate etiologies of biliary duct changes. An adequate liver biopsy i.e., at least 1.5 cm, and the presence of at least 10 complete portal tracts were considered. Patients with other etiologies of portal vein thrombosis or biliopathy such as hepatic and pancreatic malignancies, chronic liver disease, sclerosing cholangitis, and biliary calculi were excluded. Ten cases were excluded as per the mentioned exclusion criteria.

The institutional ethics committee of the Institute of Liver and Biliary Sciences, Delhi approved the study (approval code: IEC/2020/82/MA11B on 11th October 2020).

### 2.2. Methods

Demographic details and laboratory tests were recorded from the electronic medical files.

#### 2.2.1. Histopathology

Liver biopsies were retrieved and analyzed by two hepatopathologists (AR and CB) in a blinded manner and further discussed over a multi-head microscope to conclude if there are any discrepancies.

#### 2.2.2. Histopathological Characteristics Assessed

Following histological features were assessed and semi-quantitatively graded for each of the 50 liver biopsies: biopsy length, number of portal tracts (PT), type and density of PT inflammation, bile ductular reaction, any atypia of bile ducts, presence of cholestasis, presence of portal vein thickening, thrombosis, narrowing or obliteration, presence of dilated branches of the portal vein, presence of aberrant vascular channels around PTs, sinusoidal dilatation and congestion, central vein dilatation, remnant PTs, any parenchymal changes, and fibrosis extent including periportal and perisinusoidal fibrosis and cirrhosis.

#### 2.2.3. Statistical Analysis

All data collected were analyzed using IBM Statistical Package for Social Sciences software (SPSS version 22, IBM Corp., Armonk, NY, USA). The continuous data were represented as mean ± SD or median (interquartile range) and categorical as percentages or in frequency as appropriate. Categorical data were analyzed using the Chi-square test or Fisher’s exact test as applicable. *p*-value < 0.05 is considered significant.

## 3. Results

A total of 50 patients with portal cavernoma cholangiopathy were included in the study. The mean age of the patients was 26.2 ± 11.6 years. There was a male preponderance with male to female ratio of 2.6:1. Basic parameters are summarized in Table 1. Imaging characteristics were as follows: bilobar intrahepatic biliary radical dilatation in 98% and common bile duct abnormality in 100% of patients (Figure 1a, b). Extrinsic bile duct impressions were present in 77% (Figure 1c–e), stricture in 16% (Figure 1f, g), and smooth contour in 6% was observed. All strictures were <10 mm in length. Radiological findings are summarized in Table 2. Shunt surgeries were done in 52 patients, 6 underwent splenectomy and 2 cholecystectomies. Liver function tests were mild to moderately deranged with median levels of total bilirubin 1.5 mg/dL (IQR 0.8–2.4), SAP 109.5 IU/L (IQR 70–193), and GGT 25.5 IU/L (IQR 14.7–74).

### Histopathology

Table 3 describes the histopathological characteristics and their prevalence in portal biliopathy (*n* = 50).

The mean biopsy length was 1.67 ± 0.28 cm with a number of portal tracts of 9.72 ± 3.04. The liver architecture was maintained in the majority of the cases. Histopathological findings in PCC cases are described in Table 3. Portal tracts displayed minimal to mild mononuclear cell inflammation (82%) (Figure 2a). Histological characteristics of EHPVO were noted in the biopsies: portal sclerosis with portal vein phlebosclerosis in 32% (Figure 2b), dilated multiple portal venous channels (72%) (Figure 2c), hepatic artery thickening (70%) (Figure 2d), presence of aberrant vascular channels around portal tracts (54%) (Figure 2e), and elastosis of portal veins (Figure 2f). Remnant portal tracts were found in 20% of the biopsies, and Central vein and sinusoidal dilatation were also recorded in 44% and 26% of the cases respectively. 

Histological features favoring cholangiopathy were found in less than half of the cases and features were of mild grade- bile ductular reaction in 44% (Figure 3a) whereas reactive biliary changes were noted in only 10% of the cases. Hepatocellular and canalicular bile was found in only 9 cases (18%) (Figure 3b). There was increased periportal and/or zone 3 perisinusoidal fibrosis in 38% (Figure 3c). 12% of patients showed the presence of occasional thin fibrous bridges. Cirrhosis or advanced fibrosis was not seen in any of these cases. One-fourth of the cases (12) showed concomitant minimal to mild hepatocyte steatosis. 

## 4. Discussion

Portal cavernoma cholangiopathy is a type of biliary change that occurs in association with a portal vein thrombus or cavernoma. It is estimated that more than half of portal hypertension cases in developing countries can be attributed to extrahepatic portal vein obstruction, and it is one of the common causes of gastrointestinal bleeding among young patients [18].

Portal biliopathy as a clinical entity was established by a relatively small case series [5,6]. Sarin et al. [13] detected similar changes in 80% of the patients with EHPVO [2]. Since then, several investigators have reported on case series of patients with portal biliopathy. Diagnostic Imaging modalities include ultrasound, endoscopic Doppler ultrasound, Multi-detector CT, MRCP, and ERCP. A raised serum bilirubin level with a predominant increase in its direct component and an elevated serum alkaline phosphatase is an indication for performing biliary imaging in EHPVO [6].

In normal conditions, the venous drainage of the bile duct is divided into two special plexuses. An epicholedochal plexus (of Saint) forms a reticular network of veins on the outer surface of the bile ducts [19]. The paricholedochal network of Petren courses parallel to the CBD and is connected to the gastric, pancreaticoduodenal, and portal veins below, and the liver above. Its conversion into collateral veins causing pressure and bulging of the thin and flexible bile duct walls is called portal biliopathy [20]. These collaterals form portal cavernoma comprising a dense vascular pattern and fibrous stroma in the peripancreatic region along the occluded portal vein and provide an alternative route around the thrombosed segment of the portal vein.

In addition to the extrinsic compression by pericholecystic and epicholedochal [19,20], several other inter-related mechanisms are proposed for the occurrence of biliopathy: venous plexuses that expand and compress in an attempt to decompress the venous blockade [3]; the formation of new vessels and connective tissue resulting in solid tissue around the ducts [21]; extension of the thrombotic process to small venules of the bile ducts leading to ischemic injury to the bile duct, compression of the bile ducts, ischemia, or infection [22]. Such mechanisms ultimately lead to stasis, cholangitis, choledocholithiasis, and stricture formation [3].

In comparison to the published literature, the present study encompasses a relatively large cohort of patients. Although the histological changes are non-specific; however, indicative of propagated microthrombi creating chronic portal changes. Approximately 40% of cases with biliary changes and 12% of cases with thin fibrous bridges which otherwise would have been missed. There is a dearth of published literature on liver histopathology; there are only scarce case reports and case series reported [7,15]. A recent study by Pittman et al described histologic findings in liver specimens from three patients with radiologically confirmed portal cavernoma cholangiopathy [7]. A liver biopsy was performed because of clinical suspicion of a mass lesion or cirrhosis. Major histologic features were similar to our findings and included: portal venous abnormalities- portal veins obliterated/ narrowing, obstructive biliary changes- ductular reaction, and reactive epithelial atypia accompanied by a mixed inflammatory cell infiltrate. In another study, two of the three cases reported had liver biopsies done [15]. Biopsies were non-cirrhotic and either were morphologically normal or showed non-specific changes with portal fibrosis [15]. Another report of a 63 year old man with EHPVO underwent a transjugular liver biopsy that showed mild non-specific inflammation and no fibrosis [23]. 

This study has a few limitations, it is a retrospective descriptive study lacking mechanistic insights. Nonetheless, this study provides a useful addition to the literature, regarding the histological changes and their spectrum that can be noted in portal cavernoma cholangiopathy. 

## 5. Conclusions

Patients with portal cavernoma cholangiopathy showed a variable combination of portal vein abnormalities, chronic portal vascular changes, and biliary changes in the liver biopsies. The subtlety of morphological findings and a few cases with biliary changes and increased fibrosis highlights the histopathological spectrum that can be noted. 

## Figures and Tables

**Figure 1 diagnostics-13-00436-f001:**
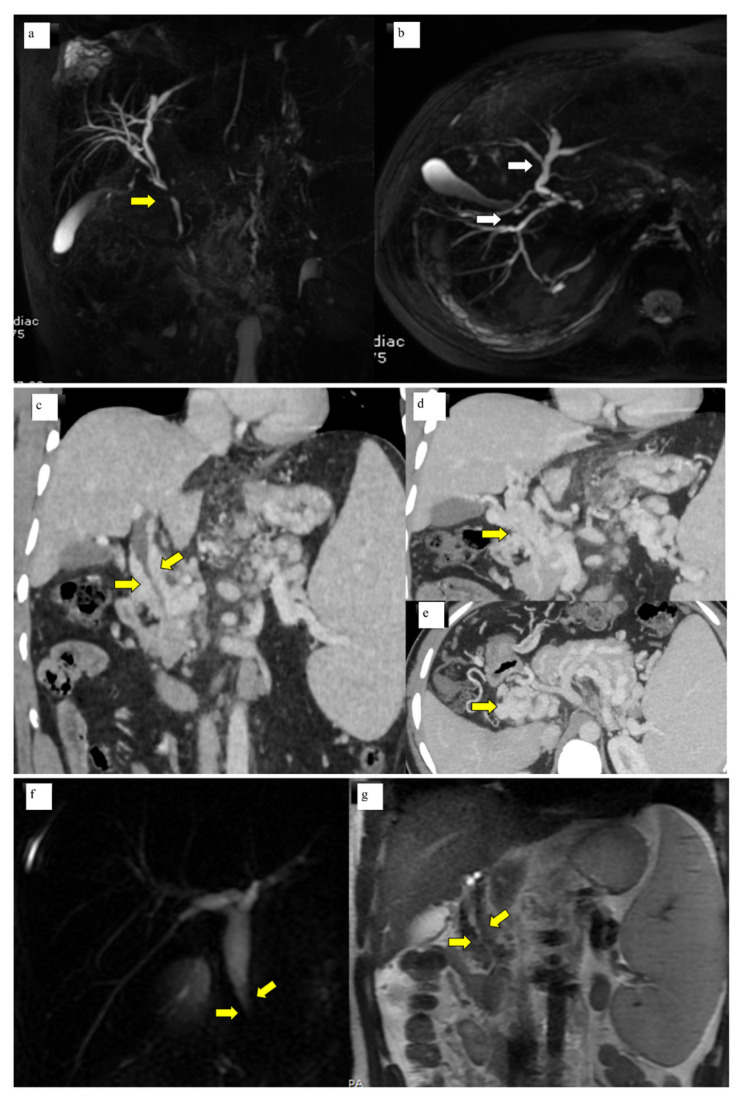
(**a**) MRCP Coronal & (**b**) 3D axial MRCP showing extrahepatic (Bold yellow arrow) and intrahepatic biliary dilatation (bold white arrows) with ductal wall irregularity in a case of well-delineated portal biliopathy of a 16-year-old young boy. (**c**) Coronal Contrast-enhanced CT scan of the abdomen showing opacified paracholedochal collaterals (bold yellow arrows) along the length of the common bile duct causing extrinsic luminal compression. (*d*) Coronal Contrast-enhanced CT Scan of the abdomen showing multiple opacified portal cavernoma collaterals along the entire length of the common duct (bold yellow arrow), extending up to suprapancreatic segment. (**e**) Axial contrast-enhanced CT section of the upper abdomen shows portal cavernoma as a bunch of collaterals in the spleno-portal confluence and the SMV which cannot be separately identified from the collaterals (yellow bold arrow). (**f**) Coronal 2D Magnetic resonance cholangiopancreatography (MRCP) image showing narrowing of the distal common duct segment (bold yellow arrows) due to extra luminal collaterals seen as T2 hypointense flow voids (bold yellow arrows). (**g**) Coronal T2 WI of the MRI of the upper abdomen.

**Figure 2 diagnostics-13-00436-f002:**
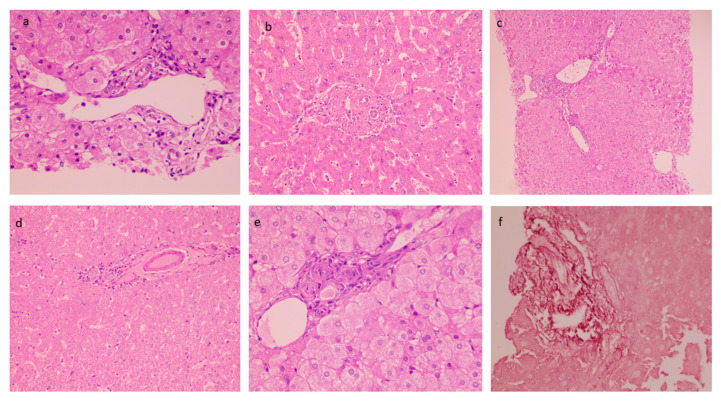
Histological characteristics of EHPVO: (**a**) Minimal Portal Inflammation (20×, HE); (**b**) portal sclerosis (20×, HE); (**c**) dilated multiple portal venous channels (20×, HE); (**d**) hepatic artery thickening (10×, HE); (**e**) presence of aberrant vascular channels around portal tracts (20×, HE); and (**f**) elastosis of portal veins (20×, orcein).

**Figure 3 diagnostics-13-00436-f003:**
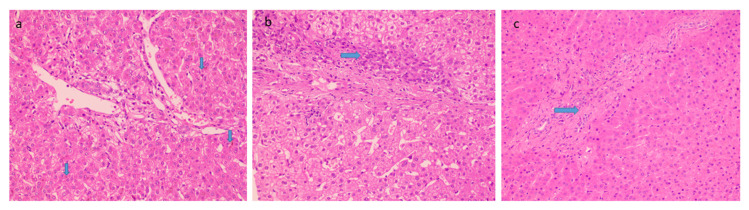
Histological features favoring biliopathy (**a**) bile stasis (20×, HE, arrow) (**b**) ductular reaction (20×, HE, arrow) and (**c**) increased periportal fibrosis (20×, HE, arrow).

**Table 1 diagnostics-13-00436-t001:** Basic attributes in portal cavernoma cholangiopathy.

Parameters	Values
Age, years	27 (15–35)
Gender, M:F	36:14
Hemoglobin, g/dL	10.8 (9–12)
Total leukococyte count/L × 10^−6^	6 (4.6–8)
Platelet count/L × 10^−9^	89 (68–118)
Serum Bilirubin, mg/dL	1.5 (0.8–2.4)
Serum alkaline phosphatase, IU/L	109 (70–193)
Gamma Gultamyl transpeptidase, IU/L	25.5 (14.7–74)
Coagulation abnormalities, None/JAK2/MTHFR/Factor V leidon/Protein C/S deficient	41/1/4/2/2
Spleen size, cm	17 (14.7–22)
Varices, no/yes	10/40
Ascites, no/yes	46/4

**Table 2 diagnostics-13-00436-t002:** Radiological features in cases of portal cavernoma cholangiopathy.

Radiological Features	Frequencies
Thrombus length, mm	23 (10–45)
Thrombus location, along with portal vein-Splenic Vein/Superior mesenteric vein/both	17/3/10
Portal Cavernoma Cholangiopathy, classification 1/2/3a/3b	8/9/5/28
Associated Intrahepatic cavernoma, no/yes	11/39
Portosystemic collaterals, no/yes	8/42
Bilobar intrahepatic biliary radical dilatation, none/mild to moderate/severe	24/10/16
Common bile duct contours smooth/wavy or mild indentation/stricture	5/45/1
Common bile duct calculi, no/yes	41/9
Common bile duct angle, degree	140 (130–157)
Extrahepatic Common bile duct stricture, no/yes	42/8
Shunting-Proximal Lienorenal Shunt/mesocaval shunt/gastrorenal shunt	14/34/1/1

**Table 3 diagnostics-13-00436-t003:** Histopathological characteristics and their prevalence in portal biliopathy (*n* = 50).

Histopathology Parameters	Mean ± SD
Mean biopsy length (cm)	1.67 ± 0.28
The average number of portal tracts	9.72 ± 3.04
Portal mononuclear inflammation	41 (82%)
Portal mixed inflammation	9 (18%)
Bile ductular reaction; of which mild ductular reaction	22 (44%) 19 (86%)
Reactive biliary changes	5 (10%)
Portal vein narrowing or obliteration	16 (32%)
Arterialized vein wall with elastosis	3 (6%)
Recanalized dilated multiple venous channels replacing portal vein	36 (72%)
Para-portal dilated veins and sinusoids	27 (54%)
Hepatic arteries—thickened wall and increased number of profiles	35 (70%)
Increased periportal and/or zone 3 perisinusoidal fibrosis	19 (38%)
Occasional thin fibrous septal fibrosis—none of the cases developed cirrhosis	6 (12%)
Sinusoidal dilatation	13 (26%)
Central vein dilatation	20 (40%)
Remnant portal tracts	10 (20%)
Hepato-canalicular bile	9 (18%)
Concomitant minimal to mild steatosis	12 (24%)

## Data Availability

Data is available on request.

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
