# Peer review of "Histological Changes in Portal Cavernoma Cholangiopathy"

_diagnostics, 2023, doi:10.3390/diagnostics13030436_

Round 1
Reviewer 1 Report
Thank you for the opportunity to review this research paper, which covers an important issue. I realize it is difficult to make one's point in a research paper when your work is very extensive. I found this research paper very hard to follow, and at times with unsubstantiated facts. I apologize if my feedback seems harsh.
The title and other parts of the manuscript have a lot of typographical and grammatical mistakes. Please use Grammarly/Ginger to know common mistakes.
The experimental work is extensive but did not explain the results adequately.
Explain why you have conducted this research and what are the benefits of conducting this research.
The statistical analysis needs a little attention. Kindly explain the statistical tools and the outcome properly.
Author Response
Reviewer-1
Thank you for the opportunity to review this research paper, which covers an important issue. I realize it is difficult to make one's point in a research paper when your work is very extensive. I found this research paper very hard to follow, and at times with unsubstantiated facts. I apologize if my feedback seems harsh.
Response: Thank you for the suggestion, we have revised the manuscript for clarity
The title and other parts of the manuscript have a lot of typographical and grammatical mistakes. Please use Grammarly/Ginger to know common mistakes.
The experimental work is extensive but did not explain the results adequately.
Response: Thank you for the guidance and we have changed the title as per the results and revised the manuscript and checked the grammar using grammarly.
Explain why you have conducted this research and what the benefits of conducting this research are.
Response: Shunt surgeries are performed in specialized tertiary care centres. A significant cohort of cases motivated us to analyse the histology in details, where only scarce literature is available. Knowledge of spectrum of histological changes is could be an important addition to the literature, for the pathologists and clinicians reference.
The statistical analysis needs a little attention. Kindly explain the statistical tools and the outcome properly.
Response: Thank you for the comment and we have added the para for the same.
Reviewer 2 Report
The authors detail a retrospective study on pathological aspects of EHPVO. My concerns are as follows.
Major:
1. This is an area with very confusion nomenclature. Please be consistent in the use of terms and please clarify the exact target of this study. The abstract says the study is on “patients clinically diagnosed with EHPVO with portal cavernoma, cholangiopathy OR biliopathy.” So is portal cavernoma considered unnecessary for inclusion in this study? The Results section of the abstract says “patients with cavernoma AND biliopathy”. Inclusion criteria in methods says “primary portal vein thrombosis with overlying features of portal biliopathy, and an adequate liver biopsy” without mention of cavernoma. Please be consistent and please provide a precise definition (in the discussion, the authors state: The triad that needs to be fulfilled to make a diagnosis is presence of a portal cavernoma, cholangiographic changes on ERCP or MRCP, and absence of alternate etiologies of biliary duct changes. Maybe this should be the inclusion criteria?). With respect to nomenclature, also note the 2014 consensus statement suggesting the term “portal cavernoma cholangiopathy”. http://dx.doi.org/10.1016/j.jceh.2014.02.003
2. Methods: Please provide basic details. Was this a single-center study? What was the time period covered (it only says “8-year period”: please provide dates)?
3. Was this study approved by an ethics committee? If so, please state.
4. Results: Please provide more information on the baseline clinical characteristics of the 50 subjects in a Table, including BMI (if available), socioeconomic status (if available), length and/or location of thrombosis, coagulation abnormalities, congenital abnormalities, radiological features, etc. Radiological features may warrant an additional table (how many patients received MRCP, ERCP, etc.; presence of varices, splenomegaly, etc.).
5. Results: “Liver function tests were normal to slightly deranged with median levels of total bilirubin 1.5 mg/dl (IQR 0.2-34.2)” (line 117): the upper quartile has total bilirubin over 34.2 mg/dl? That is extremely high and not “slightly deranged”. Please also check your figures because this seems too high.
6. Table 1: The denominator is 50 for all items. To confirm, does this mean that liver biopsies were adequate to evaluate every single one of these items for all 50 patients on liver biopsy? Otherwise, the denominators should be adjusted and a footnote should be provided.
7. “It is estimated that more than half of portal hypertension cases in developing countries can be attributed to extrahepatic portal vein obstruction, and it is the most common cause of gastrointestinal bleeding among pediatric patients [18]” (line 175): Is this true? I read reference 18 by Sarin et al and there is no mention of these facts.
8. The discussion reviews the literature on EHPVO, but there is almost no discussion of the results of this study. Please clarify what the key findings of this study are. For example, which histological parameters appear specific to EHPVO relative to other diseases?
9. Please provide a paragraph on the limitations of this study.
10. I cannot understand the Conclusion. There are only general statements that have nothing to do with the 50 patients examined.
Minor:
1. Please check for inappropriate capitalization, incomplete sentences, subject-verb agreement, and missing punctuation.
2. IQR (line 31), IHBR (line 113), CBD (line 114): please define abbreviations at first use.
Author Response
Reviewer-2
- This is an area with very confusion nomenclature. Please be consistent in the use of terms and please clarify the exact target of this study. The abstract says the study is on “patients clinically diagnosed with EHPVO with portal cavernoma, cholangiopathy OR biliopathy.” So is portal cavernoma considered unnecessary for inclusion in this study? The Results section of the abstract says “patients with cavernoma AND biliopathy”. Inclusion criteria in methods says “primary portal vein thrombosis with overlying features of portal biliopathy, and an adequate liver biopsy” without mention of cavernoma. Please be consistent and please provide a precise definition (in the discussion, the authors state: The triad that needs to be fulfilled to make a diagnosis is presence of a portal cavernoma, cholangiographic changes on ERCP or MRCP, and absence of alternate etiologies of biliary duct changes. Maybe this should be the inclusion criteria?). With respect to nomenclature, also note the 2014 consensus statement suggesting the term “portal cavernoma cholangiopathy”. http://dx.doi.org/10.1016/j.jceh.2014.02.003.
Response- As per your suggestion and throughout the manuscript we have changed as “portal cavernoma cholangiopathy” as suggested and triad has been incorporated in the inclusion.
- Methods: Please provide basic details. Was this a single-center study? What was the time period covered (it only says “8-year period”: please provide dates)?
Response: We have incorporated the points in revised draft.
- Was this study approved by an ethics committee? If so, please state.
Response: Thank you for this point and we have added the ethical statement.
- Results: Please provide more information on the baseline clinical characteristics of the 50 subjects in a Table, including BMI (if available), socioeconomic status (if available), length and/or location of thrombosis, coagulation abnormalities, congenital abnormalities, radiological features, etc. Radiological features may warrant an additional table (how many patients received MRCP, ERCP, etc.; presence of varices, splenomegaly, etc.).
Response: Thank you for the important point, we have incorporated the baseline details in Table-1 with available information and radiological findings summarized in Table-2 as per your suggestion.
- Results: “Liver function tests were normal to slightly deranged with median levels of total bilirubin 1.5 mg/dl (IQR 0.2-34.2)” (line 117): the upper quartile has total bilirubin over 34.2 mg/dl? That is extremely high and not “slightly deranged”. Please also check your figures because this seems too high.
Response: We are sorry as the range has been mislabelled by us as IQR and we have corrected the error.
- Table 1: The denominator is 50 for all items. To confirm, does this mean that liver biopsies were adequate to evaluate every single one of these items for all 50 patients on liver biopsy? Otherwise, the denominators should be adjusted and a footnote should be provided.
Response: It is to submit that denominator is 50 for all the items given in table (now becomes table-3 as two additional tables included as per your suggestion)
- “It is estimated that more than half of portal hypertension cases in developing countries can be attributed to extrahepatic portal vein obstruction, and it is the most common cause of gastrointestinal bleeding among pediatric patients [18]” (line 175): Is this true? I read reference 18 by Sarin et al and there is no mention of these facts.
Response: Thank you for the pointing out our mistake, we have corrected this point.
- The discussion reviews the literature on EHPVO, but there is almost no discussion of the results of this study. Please clarify what the key findings of this study are. For example, which histological parameters appear specific to EHPVO relative to other diseases?
Response: We have redrafted the discussion as per your suggestion and added the histological findings discussion in the manuscript.
- Please provide a paragraph on the limitations of this study.
Response: Thank you for the suggestion, we have added the limitations of this study.
- I cannot understand the Conclusion. There are only general statements that have nothing to do with the 50 patients examined.
Response: Thank you for the comment and we have rewritten the conclusion of the study as:
“Patients with portal cavernoma cholangiopathy showed a variable combination of portal vein abnormalities, chronic portal vascular changes, and biliary changes in the liver biopsies. The subtlety of morphological findings and few cases with biliary changes and increased fibrosis highlights the histopathological spectrum that can be noted.”
Round 2
Reviewer 2 Report
The authors have significantly improved the manuscript based on reviewers' suggestions. I believe it is suitable for publication (after minor language editing) and have no further comments.